# Identification and Purification of Novel Low-Molecular-Weight Lupine Allergens as Components for Personalized Diagnostics

**DOI:** 10.3390/nu13020409

**Published:** 2021-01-28

**Authors:** Uta Jappe, Arabella Karstedt, Daniela Warneke, Saskia Hellmig, Marisa Böttger, Friedrich W. Riffelmann, Regina Treudler, Lars Lange, Susanne Abraham, Sabine Dölle-Bierke, Margitta Worm, Nicola Wagner, Franziska Ruëff, Gerald Reese, André C. Knulst, Wolf-Meinhard Becker

**Affiliations:** 1Division of Clinical and Molecular Allergology, Research Center Borstel, Airway Research Center North (ARCN), German Center for Lung Research, 23845 Borstel, Germany; AKarstedt@web.de (A.K.); dwarneke@fz-borstel.de (D.W.); saskia.hellmig@gmail.com (S.H.); mboettger@fz-borstel.de (M.B.); wolfmeinhard@aol.com (W.-M.B.); 2Interdisciplinary Allergy Outpatient Clinic, Department of Internal Medicine and Pneumology, University of Lübeck, 23538 Lübeck, Germany; 3Department of Allergology, Fachkrankenhaus Kloster Grafschaft, 57392 Schmallenberg, Germany; F.Riffelmann@fkkg.de; 4Department of Dermatology, Venerology and Allergology, Universität Leipzig, 04103 Leipzig, Germany; Regina.Treudler@medizin.uni-leipzig.de; 5Department of Pediatrics, St. Marien-Hospital, 53115 Bonn, Germany; Lars.Lange@gfo-kliniken-bonn.de; 6Department of Dermatology, University Allergy Center, Medical Faculty Carl Gustav Carus, Technische Universität Dresden, 01307 Dresden, Germany; Susanne.Abraham@uniklinikum-dresden.de; 7Division of Allergy and Immunology, Department of Dermatology, Venerology and Allergology, Charité—Universitätsmedizin Berlin, Corporate Member of Freie Universität Berlin, Humboldt-Universität zu Berlin, and Berlin Institute of Health, 10117 Berlin, Germany; sabine.doelle@charite.de (S.D.-B.); margitta.worm@charite.de (M.W.); 8Department of Dermatology, University Hospital Erlangen, Friedrich-Alexander-Universität Erlangen-Nürnberg (FAU), 91054 Erlangen, Germany; Nicola.Wagner@uk-erlangen.de; 9Department of Dermatology and Allergology, Klinikum der Universität München, 80337 Munich, Germany; Franziska.Rueff@med.uni-muenchen.de; 10Allergopharma, 21465 Reinbeck, Germany; gerald.reese@allergopharma.com; 11Department of Dermatology/Allergology, University Medical Center Utrecht, Utrecht University, 3508 GA Utrecht, The Netherlands; A.C.Knulst@UmcUtrecht.nl

**Keywords:** cross-reaction, flour, food allergy, individualized diagnostics, legumes, lupine, lipid transfer protein, peanut, profilin

## Abstract

Lupine flour is a valuable food due to its favorable nutritional properties. In spite of its allergenic potential, its use is increasing. Three lupine species, *Lupinus angustifolius*, *L. luteus*, and *L*. *albus* are relevant for human nutrition. The aim of this study is to clarify whether the species differ with regard to their allergen composition and whether anaphylaxis marker allergens could be identified in lupine. Patients with the following characteristics were included: lupine allergy, suspected lupine allergy, lupine sensitization only, and peanut allergy. Lupine sensitization was detected via CAP-FEIA (ImmunoCAP) and skin prick test. Protein, DNA and expressed sequence tag (EST) databases were queried for lupine proteins homologous to already known legume allergens. Different extraction methods applied on seeds from all species were examined by SDS-PAGE and screened by immunoblotting for IgE-binding proteins. The extracts underwent different and successive chromatography methods. Low-molecular-weight components were purified and investigated for IgE-reactivity. Proteomics revealed a molecular diversity of the three species, which was confirmed when investigated for IgE-reactivity. Three new allergens, *L. albus* profilin, *L. angustifolius* and *L*. *luteus* lipid transfer protein (LTP), were identified. LTP as a potential marker allergen for severity is a valuable additional candidate for molecular allergy diagnostic tests.

## 1. Introduction

Flour from raw lupine seed is used increasingly as a protein source in Australia, New Zealand, The USA and European countries, where lupine serves as a replacement for animal proteins, i.e., milk, egg white, and potentially genetically modified soy products [1]. Since lupine lacks gluten, lupine-containing products are recommended for patients with wheat protein allergy and coeliac disease. Further beneficial effects are increased satiety, reduced energy intake, hypolipidemia, and a decrease in blood glucose concentration [2]. In addition, lupine is an important protein source in the vegan diet, which is experiencing a growing interest. The genus *Lupinus* belongs to the *Papilionaceae* subfamily in the *Leguminosae* family, which among others, also contains peanuts, soybean, beans, peas, chickpeas, lentils, and fenugreek. However, the dietary value of lupine proteins is higher than that of beans or peas, which is mainly due to high concentrations of the essential amino acids lysine, leucine and threonine, which are higher only in soybeans. It has relatively high concentrations of protein and dietary fiber in contrast to digestible carbohydrates and lipids (summarized in [3]). Three lupine species (*Lupinus albus*, *L. luteus*, and *L. angustifolius*) are used as food as well as a food additive to fortify wheat flour, which may contain up to 10 or even 15% of lupine flour.

In spite of its nutritional value, lupine is an upcoming food allergen responsible for severe food allergy. Since the first publication of lupine allergy in a peanut-allergic child [4] and the systematic review of 151 cases in 2010 [3], further cases were reported worldwide. It is, therefore, interesting that detailed knowledge on individual lupine allergens is still sparse and that in vitro allergy diagnostic tests still rely on lupine seed extract only (*Lupinus albus,* ImmunoCAP f335, Thermo Fisher Scientific [5]) [3]. This is in contrast to other relevant allergenic legumes (peanut and soy), for which, besides the whole extract, individual allergens covering relevant protein families and marker allergens are available for routine allergy diagnostic tests. As has been shown for other whole allergen extracts before, it is highly probable that the only available lupine extract lacks some relevant allergens and, in addition, does not satisfactorily address species-specific differences. In addition, most patients who are confronted with the question of food allergy to lupine do not even know that it is a food as well as a food ingredient. Therefore, it can be safely assumed that not all lupine-allergic patients are diagnosed correctly. Most clinical studies on lupine allergy vary with regard to design, population, geographic origin, and endpoints. In general, there are three variants of lupine allergy: primary food allergy, secondary (pollen-associated) food allergy, and occupational inhalant allergy with or without associated intolerance of lupine ingestion [3]. Depending on the selection of the study population, the percentage of clinically relevant (symptoms of allergy after ingestion or inhalation) and non-relevant lupine sensitization (positive IgE-antibody detection only) differs. With regard to the data published so far, cross-reactivity between lupine and soybean, beans, lentils, and peas does not seem to be of much clinical relevance when compared with the cross-reactivity between lupine and peanut (summarized in [3,6]). The study of Moneret-Vautrin and co-authors in 1999 had shown that 28% of peanut-allergic patients also experienced symptoms after lupine ingestion [7], in a study from Peeters and co-authors in 2009, the percentage of lupine-allergy in peanut-allergic patients was 35% [1]. None of these patients had been aware of their allergy to lupine. This unawareness is a considerable problem to the present day [8], indicating that peanut-allergic patients are at risk of reacting to lupine-containing food with severe allergic symptoms. Those patients will definitely avoid peanut, but not necessarily lupine, when they are oblivious to the phenomenon of cross-reactivity. In order to understand cross-reactivity and the fact that raw lupine in the form of seed, flour or dust is known to induce different disease entities (food allergy as well as occupational allergy like baker’s asthma) [9], several investigations focus on molecular allergology trying to identify and characterize single lupine allergens (summarized in [3], updated in [6]). Regarding the identification of clinically relevant single allergens in lupine species, the storage proteins were the first to be studied, as they are in other clinically relevant legumes associated with severe allergic reactions. IgE-reactivity was demonstrated for the δ-conglutin (9 kDa and 4 kDa), non-reduced γ-conglutin or α-conglutin with a molecular weight of 43 kDa (glycinin or legumin) (summarized in [3]). The precursor of the β-conglutin of *Lupinus angustifolius* (20 kDa to 80 kDa), which is a vicilin-like storage protein, was the first accepted by the WHO/International Union of Immunological Societies (IUIS) Allergen Nomenclature Subcommittee as allergen Lup an 1 [10]. However, since it is not generally documented by the food producers, which lupine species has been used for flour in foods, and because the phenomenon of clinically relevant cross-reactivity has not been fully understood, it is necessary that investigations for species-specific differences should become possible in the future. Based on a German multicenter study, patients allergic to lupine with and without cross-reactivity to peanut were included, their clinical data recorded, and their sera investigated for sensitization profiles with the three lupine species, revealing species-relevant differences. Subsequently, the sera of these patients were used for the identification of new single lupine allergens, which were purified (lipid transfer protein) and recombinantly produced (profilin) and applied in IgE-detection measures.

## 2. Materials and Methods

### 2.1. Study Group

A total of 31 individuals have been included: 5 patients with lupine allergy alone (including one with strongly suspected allergy to lupine-containing foods), 10 patients with peanut allergy and lupine allergy (including four with strongly suspected lupine allergy), 11 patients with peanut allergy and lupine sensitization, two patients with peanut allergy without proven lupine sensitization, two patients with lupine and peanut sensitization only, and one non-allergic individual, whose serum served as negative control were recruited during clinical work in the allergy outpatient clinics of Borstel and Lübeck, Germany, as well as in study centers in Schmallenberg, Leipzig, Bonn, Dresden, Berlin, Erlangen, and Munich, Germany, on an ongoing basis. In addition, the University of Utrecht participated in this investigation. The patients were characterized by standardized questionnaires and specified medical history, and the sensitization to lupine and peanut was investigated via ImmunoCAP (ImmunoCAP, Phadia AB, Freiburg, Germany, and Uppsala, Sweden) (Table 1). In cases without convincing clinical history for anaphylaxis, an open oral food challenge was performed with lupine flour [11] in some centers. Three had undergone double-blind placebo-controlled lupine challenge in Utrecht with positive results and were included in this study [1]. The study was approved by the ethics committee of the University of Lübeck, Germany, with approval numbers 10-124, 10-126, and 13-086. All patients gave informed consent.

### 2.2. Lupine Extract Production and Protein Identification According to the Molecular Weight

Lupine extracts were produced from dry seeds of *Lupinus angustifolius*, var. Boregine (Saatszucht Steinach, Steinach, Germany), *Lupinus albus*, var. Feodora (Saaten-Union GmbH, Isernhagen, Germany), and *Lupinus luteus*, var. Juno ZS (Feldsaaten Freudenberger GmbH and Co. KG, Krefeld, Germany). For the production of the protein extracts, the lupine seeds were briefly frozen with liquid nitrogen, then ground in a coffee grinder to obtain flour, and afterwards extracted at different pH-levels.

In order to identify as many allergens as possible, we performed two extraction protocols, acidic and alkaline protein extraction, of the three relevant lupine species.

#### 2.2.1. Alkaline Extraction

Samples of 2 × 4 g lupine flour were dissolved each in 40 mL of 0.2 M ammonium hydrogen carbonate (NH_4_HCO_3_, pH 8.0) and incubated for 30 min at 37 °C on a shaker. The flour was then centrifuged for 30 min at 13,000× *g*. Afterwards, the dialysis of the supernatant against Milli-Q water was performed with a dialysis tube, cutoff 3 kDa. The solution was again centrifuged at 13,000× *g*, and the supernatant filtered with a 0.45 µm pre-syringe filter (membrane polyethersulfone (PES)).

#### 2.2.2. Acidic Extraction

2 × 4 g lupine flour was dissolved each in 40 mL of 0.1 M ammonium acetate (CH_3_COONH_4_, pH 5.0) and incubated for 6 h at 4 °C on a shaker. The flour was then centrifuged for 30 min at 13,000× *g*, and the supernatant dialyzed against Milli-Q water. The Milli-Q water was changed twice during this process. The solution was subsequently centrifuged again at 13,000× *g* for 20 min, and the supernatant filtered with a 0.45 µm pre-filter for syringes.

#### 2.2.3. Gel Electrophoresis of Different Lupine Extracts

The separation of proteins of a molecular weight between 14 kDa and 100 kDa was performed via SDS-PAGE according to Laemmli UK (1970) [12].

Depending on the required volume, the samples were taken up 1:2 in 2-fold reducing sample buffer (200 mM Tris/HCl; 2 mM EDTA; 2% SDS; 25% glycerin; 1% dithiothreitol (DTT); 0.02% bromophenol blue, pH 6.8), or 1:5 in 5-fold reducing sample buffer (500 mM Tris/HCl; 5 mM EDTA; 5% SDS; 25% glycerin; 2.5% DTT; 0.02% bromophenol blue (pH 6.8)) and boiled for 5 min at 95 °C. Unless otherwise noted in the results section, 40 µg protein per cm gel were used for SDS-PAGE.

For better separation of proteins in the low molecular range, NuPAGE Bis-Tris 4–12% and NuPAGE Bis-Tris 12% ready-to-use gels (Invitrogen, Carlsbad, CA, USA) were used.

In combination with the 2-(N-morpholino)ethanesulfonic acid (MES) buffer (50 mM MES, 50 mM Tris, 1 mM EDTA, 0.1% SDS), these gels provide a high-resolution separation of the proteins in the low molecular range (<40 kDa).

The proteins in the acidic extracts of *L. angustifolius* and *L. luteus* were further separated by gel filtration (size exclusion chromatography (Superdex 75)) and ion-exchange chromatography (source Q) (see below).

#### 2.2.4. In Silico Analysis

Protein, DNA and EST databases were queried for lupine proteins homologous to already known legume allergens.

IgE-reactive low-molecular-weight (LMW) proteins were further investigated by N-terminal sequencing and mass spectrometric analysis. Homology search in an expressed sequence tag (EST) database revealed a cDNA sequence (FG090100), which was used for expression in *E. coli*. The resulting recombinant protein was used for immunoblot inhibition studies.

#### 2.2.5. 2D Fluorescence Difference Gel Electrophoresis (DIGE)

Protein extracts obtained via alkaline extraction of flour from all three lupine species were studied by 1D- and 2D-SDS-PAGE by means of a Refraction-2D labeling kit (NH DyeAGNOSTICS, 1 × 1, 8 nmol PR08, Halle, Germany). The extracts were solved in Milli-Q water, concentrated in a vacuum concentrator and used in the 2D-labeling kit applying 50 µg of each species and filled up to 10 µL of a compatible buffer. The buffer was produced with Tris (30 mM), urea (7 M), thiourea (2 M), 3-[(3-cholamidopropyl)dimethylammonio]-1-propanesulfonate (CHAPS) (4%) (pH 8.5). The working solution consisted of the different dyes (G-Dye 200 (green color), G-Dye 300 (red color), prepared in 4.5 µL G-Dye solvent. The samples (50 µg each plus buffer, 10 µL) plus 1 µL working solution were briefly stirred and centrifuged. Two samples of *L. angustifolius* extracts and one *L. albus* extract were dyed green, and two *L. luteus* extracts and one *L. albus* extract dyed red. These samples were cooled for 30 min on ice, and the reaction stopped with 1 µL of G-Dye stop solution. These solutions were briefly stirred, put again on ice for 10 min and subsequently investigated or frozen at −80 °C.

The investigation in the first-dimension gel electrophoresis was performed using the labeled lupine extracts as follows: 5 µL of *L. angustifolius* extract, dyed green plus 5 µL of *L. luteus* extract labeled red. The other samples of different lupine species extracts were combined accordingly.

The samples were added to 155 µL of rehydration buffer that consisted of 8 M urea 2% CHAPS, 0.5% ampholyte (Servalyt, pH 3–10, Serva, Heidelberg, Germany), 0.002% of bromophenol blue up to a final concentration of 20 mM DTT to be added directly before use.

The three combinations of lupine extract samples that were to be compared were pipetted into a ZOOM chamber (Invitrogen, Carlsbad, CA, USA), and one Novex™ ZOOM™ IPG-strip pH 3–10 L (Thermo Fisher Scientific, Invitrogen, Carlsbad, CA, USA) each added to the chamber, and incubated overnight at room temperature. The run of proteins in the first dimension was performed at 2 W, 2 mA, 200 V for 20 min, 450 V for 15 min, 750 V for 15 min, 2000 V for 60 min. After discontinuation, the chamber was stored at −80 °C. The second-dimension run was prepared as follows: the ZOOM strips were incubated in an equilibration buffer I and II for 15 min each in a rolling incubator. Equilibration buffer consisted of 0.05 M Tris, 6 M urea, 30% glycerin, 2% SDS, 2% bromophenol blue; 5 mL of equilibration buffer I consisted additionally of 0.05 g DTT; equilibration buffer II (5 mL) consisted additionally of 0.125 g iodoacetamide. After the incubation was finished, agarose (0.5 g agarose in 100 mL MES buffer) and marker proteins were prepared by heating, the strips were added to the gel and sealed with agarose. 10 µL of marker per pocket were added, and the run was started at 50 V for 20 min, then continued at 200 V. Afterwards, the gels (4–12% Zoom Gel, Invitrogen) were conserved with 40% ethanol plus 10% acetic acid. The readout was performed via Chemidoc MP (Bio-Rad, Hercules, CA, USA), Cy 3 adjustment on the Chemidoc for dye 200 (green), Cy 5 adjustment for dye 300 (red).

### 2.3. Identification and Purification of New Single Allergens

#### 2.3.1. Recombinant Production of the *Lupinus albus* Profilin

Based on the sequence information obtained from the EST database (accession number FG090100), the company GeneArt (Regensburg, Germany) was able to identify the gene for the recombinant synthesis of profilin in *E. coli*. The gene was present in the vector pMA-T and thus obtained the antibiotic resistance to ampicillin used for selection. For expression, the ordered gene first had to be cloned. The transformation into living cells was used during cloning to propagate the DNA (plasmid).

The transformation was always performed with calcium-competent *E. coli* cells, first with TOP10F’ cells (Invitrogen, Carlsbad, CA, USA) and then BL21 DE3 cells (Stratagene, La Jolla, CA, USA).

After transformation into chemically competent BL21 DE3 cells of *E. coli* and induction with isopropyl-β-D-thiogalactopyranoside (IPTG), they produced the protein profilin.

10 mL of nutrient medium were inoculated with a clone picked from an agar plate, where *E. coli* cultures were growing, and incubated in an overnight culture at 37 °C on a shaker. The isolation of the plasmid DNA from the bacteria cells was achieved with the GeneJET Plasmid Miniprep Kit according to the manual (Thermo Fisher Scientific, Bremen, Germany) [13]. The restriction of the plasmid DNA from the bacteria cells was made with FastDigest enzymes NdeI and XhoI, according to the manual (Thermo Fisher Scientific, Vilnius, Lithuania). To analyze the restriction, the samples were separated in the agarose gel. Afterwards, the DNA was extracted from the gel according to the manufacturer’s instructions with the GeneJET gel extraction kit (Thermo Fisher Scientific, Bremen, Germany) [13].

The pET23b vector (Novagen, Darmstadt, Germany), which is required for expression, contains after the stop codon the information on the synthesis of a His-tag. The His-tag consists of six histidine residues and serves for the later isolation of the protein. For ligation, the pET23b vector was cut with the same restriction enzymes under the same conditions as the insert for ligation. The ligation of the pET23b vector with the profilin insert was made with T4-Ligase according to the manual (Thermo Fisher Scientific, Vilnius, Lithuania). The ligation sample was transformed into new cells, these multiplied in an overnight culture, and the plasmid DNA was isolated by miniprep (above). Using DNA-sequencing (MWG, Biotech AG, Ebersberg, Germany), the DNA sequence could be confirmed. This was followed by the transformation of the Plasmid DNA in BL21 DE3 expression cells.

#### 2.3.2. Expression

10 mL of preculture were infected with a clone picked from the agar plate, incubated overnight at 37 °C and transferred on the following day into 1 L nutrient medium with an additional 1 mL of ampicillin. The culture was incubated to a cell density of OD600 = 0.6–0.9. By adding IPTG (1 mM, final concentration), the protein biosynthesis of the recombinant profilin was induced. This solution was incubated for 3 h at 37 °C on a shaker and centrifuged (4000× *g*, 15 min, 4 °C). The supernatant was removed, and the pellet was processed for purification under native conditions and denaturing conditions.

#### 2.3.3. Purification under Native Conditions

The pellet was dissolved in 27 mL of lysis buffer (native) (300 mM NaCl, 50 mM Na_2_HPO_4_, 10 mM imidazole pH 8.0), 3 mL of buffer for lysis (bug buster, Merck, Darmstadt, Germany; 100 mg/mL) and 10 µL of benzonase. The bug buster served to break up the bacterial cell walls in order to facilitate the subsequent isolation of the recombinant profilin. The mixture was incubated for 1 h at 4 °C in a rolling incubator and then centrifuged (13,000× *g*, 15 min, 4 °C). This pellet was purified under denaturing conditions (see below). The supernatant was isolated via metal affinity chromatography. The supernatant was mixed with approx. 5 mL of the column material HisPur™ cobalt resin (Thermo Fisher Scientific, Rockford, IL, USA) and incubated for 1 h in a rolling incubator. Meanwhile, the His-tag of the recombinant profilin bound to the cobalt (Co^2+^) in the column material. Subsequently, the mixture was transferred to a column and the sample re-collected after the column material had settled. After the run, the first washing step was performed with 50 mL of lysis buffer (native, 300 mM NaCl, 50 mM Na_2_HPO_4_, 10 mM imidazole, pH 8.0), and a second wash step with 20 mL of wash buffer (native, 300 mM NaCl, 50 mM Na_2_HPO_4_, 40 mM imidazole pH 8.0) was carried out. The recombinant profilin was then eluted in four fractions with 5 mL of elution buffer (300 mM NaCl, 50 mM Na_2_HPO_4_, 250 mM imidazole each. Through the increased concentration of the imidazole, the elution buffer cleared the recombinant profilin from the column material. All fractions were collected and characterized by means of SDS-PAGE. The fractions, which contained the recombinant profilin, were pooled and dialyzed overnight against Milli-Q water. The protein content was then determined using the Bradford method Pierce™ Coomassie (Bradford) protein Assay Kit, Thermo Fisher Scientific, Waltham, MA, USA) and the profilin further purified via preparative SDS-PAGE (Model 491 Prep Cell, Bio-Rad Laboratories, Inc., Model 200/2.0 power supply; wide mini sub cell; Hercules, CA, USA).

#### 2.3.4. Preparative SDS-PAGE

Using preparative SDS-PAGE (prep cell model 491) from Bio-Rad (Hercules, CA, USA), the recombinant profilin, which was expressed and isolated under native conditions, was separated from further contaminants. The pooled and washed samples from the elution (above) were used in a ratio of 1:2 in a 2-fold-reducing, or 1:5 in a 5-fold-reducing sample buffer and then heated for approx. 5 min at 95 °C. For the preparative SDS-PAGE, an 11% separation gel and a 4% collection gel were used.

The column was constructed and filled with the separating gel. Then the gel was covered with water-saturated n-butanol and left to polymerize overnight. The gel was washed with Milli-Q water, and the collection gel was installed onto the separating gel. The column with the now polymerized gel was installed in the prep cell, which was filled with running buffer. The sample was applied and separated in the gel at 150 V for 80 min. The voltage was increased to 250 V for the remaining electrophoresis time.

After the bromophenol peak became visible, 95 fractions of 6 mL each were collected. The collected fractions were then investigated for the recombinant profilin.

#### 2.3.5. Separation, Isolation and Purification of Proteins via Different Chromatography Methods

The Superdex 75 column (10/300 GL, GE Healthcare, Uppsala, Sweden) is suitable for optimal separation in the range of 3000–70,000 Da.

The acidic lupine extracts were aliquoted at a protein content of 6 mg/mL each, freeze-dried and stored at −20 °C. For the Superdex 75 procedure, the extract was resuspended in 10 mL Milli-Q water and transferred to the sample loop of the chromatography system (ÄktaPurifier, GE Healthcare, Uppsala, Sweden). A buffer of 0.2 M ammonium hydrogen carbonate was used. At a flow rate of 0.5 mL/min, 25 mL per run were collected as 0.8 mL fractions. The detection of the proteins was performed at 280 nm. The individual fractions were investigated with SDS-PAGE and Coomassie staining and subsequently by immunoblot. The further natural purification of the *L. albus* profilin was not successful, which is why we promoted its recombinant production. Since the purification of the recombinant profilin via further chromatography steps (size exclusion chromatography and ion-exchange chromatography (source 15Q 4.6/100 PE, GE Healthcare, Uppsala, Sweden) were not successful, it was further purified via preparative SDS-PAGE.

### 2.4. Identification, Isolation and Purification of Natural Lupine Lipid Transfer Proteins

SDS-PAGE and immunoblotting with patients’ sera revealed LMW proteins of about 12 kDa in the acidic extracts (see above) of *L. angustifolius* and *L. luteus*. In order to isolate and purify them, 5 mg of whole lupine acidic extract per run underwent size exclusion chromatography (Superdex 75; 10/300 GL, GE Healthcare, Uppsala, Sweden) (above). In total, there were nearly 20 runs, which were performed with ammonium hydrogen carbonate buffer (0.2 M NH_4_CO_3_).

The obtained fractions underwent SDS-PAGE. Those fractions containing LMW proteins were pooled, re-buffered and concentrated by Amicon Ultra-15, (3 kDa, Millipore) and prepared for purification via ion-exchange chromatography (source 15 S, GE Healthcare, Uppsala, Sweden). The respective sample was first dialyzed three times against Milli-Q water so that the buffer that remained in the Superdex 75 was removed, and then three times against 50 mM sodium acetate buffer (pH 5.5). The run was started using samples in 100% of buffer A (50 mM sodium acetate pH 5.5). It was rinsed until the baseline was reached, then a gradient was generated. This spanned from 100% of buffer A to 100% B (50 mM sodium acetate plus 1 M sodium chloride pH 5.5) within 60 mL. Again, the obtained fractions were concentrated via Amicon Ultra-15, (3 kDa, Millipore), subsequently Superdex-peptide (10/300 GL, GE Healthcare, Uppsala, Sweden) was performed in case the result was not pure enough. SDS-PAGE of the potential *L. angustifolius* (*L. luteus*) lipid transfer protein (LTP) was then performed under reducing and nonreducing conditions. Immunoblotting revealed a single IgE-reactive band at ca. 12 kDa.

N-terminal sequencing (above) and database research was performed on the respective single allergens.

### 2.5. Immunoblotting

For blotting of the recombinant profilin, the method described above was used. For immunoblot analysis, proteins were transferred to polyvinylidene fluoride (PVDF) membranes (BIORAD, Immuno-Blot PVDF Membranes for protein blotting, BIO-RAD 0.2 µm) by semi-dry blotting for 45 min at 0.8 mA/cm^2^ as described previously [14]. Membranes were blocked for 2 h with SynBlock (Bloomington, IN, USA). The subsequently performed blots were incubated in TTBS (100 mM Tris/HCl, 100 mM NaCl, 2.5 mM MgCl_2_, 0.05% Tween-20, pH 7.4) for 30 min [15].

For immunoblot analysis, the patient sera were applied in a 1:10, 1:20 or 1:40 dilution. (For the different detection methods, see below).

### 2.6. Immunological Antigen Detection on Blotting Membranes

In our study, two different approaches to immune detections were applied:

Staining with NBT/BCIP (TBS-AP: 100 mM Tris/HCl, 100 mM NaCl, 5 mM MgCl_2_, pH 9.5). The chromogen substrate solution (nitro blue tetrazolium chloride (NBT)/5-bromine-4-chloro-3-indoxyl phosphate (BCIP)) induces an enzymatic reaction, by which the binding of the alkaline phosphatase-conjugated antibody to the protein becomes visible by means of a change in color.

Staining with horseradish peroxidase (HRP): A chemiluminescence reaction is catalyzed by an HRP-conjugated secondary antibody. The HRP catalyzes the oxidation of luminol (Bio-Rad, Hercules, CA, USA) in the presence of hydrogen peroxide (Bio-Rad, Hercules, CA, USA), and its luminescence can be detected with a digital imaging system, the ChemiDoc MP (Bio-Rad, Hercules, CA, USA). The PVDF membranes were first placed in Tris-buffered saline (TBS)-Tween buffer (pH 7.4) and incubated with gentle shaking. TBS-Tween buffer was used to block free binding sites on the blotting membrane to prevent unspecific binding.

The membrane was incubated overnight with the primary antibody, in most cases, patient serum (diluted 1:20 in TBS-Tween buffer pH 7.4). The membrane was washed three times with TBS-Tween buffer (pH 7.4) for 10 min each. Each suitable secondary antibody was incubated for 3 h while softly shaking. For detection with NBT/BCIP and alkaline phosphatase-conjugated (APC) mouse anti-human IgE antibody (dilution 1:10,000 in TBS-Tween buffer pH 7.4) (BD Pharmingen Bioscience, Heidelberg, Germany) was used. As a secondary antibody for the chemiluminescence, an HRP-conjugated mouse anti-human IgE Fc antibody (diluted 1:10,000 in Tris-Tween buffer pH 7.4 (SouthernBiotech, Birmingham, AL, USA)) was used. Free secondary antibody was removed by washing three times with TBS-Tween buffer (pH 7.4). For NBT/BCIP labeling, the membrane was washed a second time in TBS buffer (pH 9.5) to ensure optimal pH conditions for the enzymatic reaction with the chromogen substrate solution. During the washing procedure, the individual solutions, NBT and BCIP, were heated to 37 °C and then mixed with the chromogen substrate solution and applied to the membrane. The solution remained on the membrane until the proteins were clearly visible, then the reaction was stopped by transfer of the membrane into Milli-Q water. For the detection of chemiluminescence, the membrane was incubated for 5 min in 3 mL of luminol/enhancer reagent (Clarity Western ECL Substrat Bio-Rad, Hercules, CA, USA) and with 3 mL of peroxide reagent. The detection and documentation were performed with the ChemiDoc MP (Bio-Rad, Hercules, CA, USA).

### 2.7. Sequence Alignment

Sequence alignment was performed by use of BLAST (Clustal Omega, Wellcome Genome Campus, Hinxton, Cambridgeshire, CB10 1SD, UK).

### 2.8. N-Terminal Sequence Analysis

After blotting, the polyvinylidene fluoride (PVDF) membrane was washed with Milli-Q water, stained with 0.1% Coomassie in 50% methanol, destained in 50% methanol and air-dried. Protein bands were excised, and microsequencing was performed on a Procise protein sequencer with an online phenyl thiohydantoin (PTH) amino acid analyzer (PE Biosystems, Weiterstadt, Germany) [15].

### 2.9. Mass Spectrometry

The molecular masses of the protein fraction were analyzed using a high-resolution electrospray ionization (ESI) Fourier transform ion cyclotron resonance (FT ICR) mass spectrometer (Thermo Fisher Q Exactive Plus serious #387 and Advion NanoMate). For a straightforward interpretation of the heterogeneous samples, the obtained positive ion mass spectra were charge deconvoluted. Mass numbers refer to the monoisotopic mass of the neutral molecules. Tryptic mass fingerprinting was performed as described previously [16]. Briefly, Coomassie-stained protein bands were excised, destained and digested overnight with trypsin (trypsin gold, mass spectrometry grade; Promega, Mannheim, Germany) as described previously [17]. Afterwards, the corresponding tryptic fragments were mixed with 50% ACN/0.1% FA. The samples were analyzed by Thermo Fisher Q Exactive Plus serious #387 and Advion NanoMate with MS/MS. External mass calibration was performed with an appropriate mixture of peptides. Mass spectrometric data were analyzed with XCalibur Software (Thermo Fisher Scientific, Waltham, MA, USA).

## 3. Results

### 3.1. Comparison of Different Protein Extractions of Three Lupine Species

Immunoblots of the different extracts from the three lupine species that are of nutritional and commercial relevance for the community revealed differences between the species and the method of extraction (data not shown) with regard to the proteins of certain molecular weight and the respective concentration in the extract (data not shown). Investigations with sera from lupine-allergic patients revealed that seeds of different *Lupinus* species vary quantitatively in their allergen compositions. In order to investigate whether there were also qualitative differences, a 2D-DIGE was performed, including all three species, and in a subsequent experiment, peanut extract as well (Figure 1A–C).

#### 3.1.1. Immunoblot Analyses

##### Comparison of Different Protein Extractions of Three Lupine Species

The seeds of different *Lupinus* species vary qualitatively and quantitatively in their allergen compositions (Figure 2).

The patients reveal inter-individually different sensitization profiles and react differently to acidic and alkaline extractions of lupine flour. This is evident for the sera from P 3 and P 23 reacting to considerably more proteins in the alkaline *L. angustifolius* extract when compared to the acidic extraction, indicating the necessity to use more than one extraction method when searching for new single allergens. IgE-reactivity showed species-specific differences also for alkaline lupine extracts.

These results confirm on an immunological level the differences between the lupine species as had been expected based on the proteomic analysis (Figure 1B,C).

Whereas our attempt to isolate lupine storage proteins failed insofar as we could enrich only gamma conglutin (43 kDa, data not shown), but not in a sufficient amount for further complex investigations with patients’ sera, we aimed at the characterization of those proteins instead, that were identified in the low molecular range. We assumed that in these fractions, proteins of allergen families could be found, which already have some biomarker quality in other relevant food allergen sources like the closely related peanut (defensins, LTP, and profilins).

We, therefore, continued working with the acidic extracts of lupine species and more refined discrimination methods in order to identify, purify, and characterize proteins as potential allergens that were responsible for the binding of IgE in the sera of patients.

We focused on those reactions in immunoblots that were either close to a monosensitization in a patient or that stood out as prominent IgE-reactive protein bands. Since these were to be found in the low molecular range, we suspected LTP or profilin of lupine to be IgE-reactive.

### 3.2. Identification of an L. albus Profilin

Searches on sequence information of the lupine species revealed a sequence for a profilin in *L. albus* with the Accession number *FG090100.1*:

SWQTYVDEHLLCDIEGNQLTSAAIIGQDGSVWAQSSSFPQFKPEEITAIVNDFAEPGSLAPTGLYLGGTKYMVIQGEPGAVIRGKKGPGGVTVKKTNQALIIGIYDEPMTPGQCNVVVERLGDYLIDTGL

The sequence was then used to produce the recombinant profilin of *L. albus*, to confirm the sequence after expression and the molecular weight via mass spectrometry (Appendix A, Appendix A) and investigate for its allergenicity (IgE-reactivity) using patients’ sera (Figure 3A,B).

#### 3.2.1. Allergen Identification via IgE-Reactivity with Sera from Lupine-Sensitized and Lupine-Allergic Patients

Sera from lupine- and/or peanut-allergic patients showed an IgE-binding to an LMW protein of 15 kDa in an acidic extract of *L. albus*. Mass spectrometric analysis of the corresponding LMW compound provided the first evidence for the presence of *L. albus* profilin. EST database search revealed a cDNA sequence of *L. albus*, which showed more than 87% amino acid homology to the peanut (*Arachis hypogaea*) profilin Ara h 5. Since the natural profilin could not be purified in a sufficient amount via different serial purification steps, it was subsequently produced as a recombinant protein. For this, the *L. albus* cDNA sequence was used for the expression of a recombinant *L. albus* profilin. Nucleotide and protein sequences were taken from the EST sequence using the data (Appendix A) in the suppository.

Sera from lupine- and/or peanut-allergic patients who showed IgE-reactivity against the natural LMW compound also reacted with the recombinant *L. albus* profilin. This was confirmed by the experiment, where the recombinant *L. albus* profilin was able to inhibit the IgE-binding to the natural LMW compound in immunoblot analysis with the acidic *L. albus* extracts (Figure 3A,B).

#### 3.2.2. Sequence Alignment

The sequence alignment of the lupine profilin with profilin sequences from other plants with allergy relevance revealed a 92% identity with Ara h 5 (*Arachis hypogaea*, peanut profilin) and a 78% identity with Bet v 2, the profilin from birch pollen (*Betula verrucosa*). The newly identified profilin, therefore, is another component that could be responsible for cross-reactivity with peanut. In addition, the fact that patient sera are IgE reactive to lupine profilin provides some further evidence for a pollen-induced lupine sensitization.

The recombinantly produced *L. albus* profilin was consequently submitted by us to and accepted by the WHO/IUIS allergen nomenclature subcommittee as Lup a 5, isoallergen No. 0101. The molecular weight was 13.849 kDa (as deduced from the sequence), under reducing conditions (and still with the His-tag), it was 15 kDa (Table 2). An investigation for glycosylation was not performed. However, no sequence pattern for a putative N-glycosylation (NXT/S) was found in the sequence.

By tryptic mass fingerprinting and MS/MS analysis from the reduced protein sample (apparent molecular mass from SDS-PAGE: 15 kDa), the masses given in Appendix A were obtained and correspond to the calculated masses.

#### 3.2.3. *Lupinus albus* Profilin Allergenicity

*L. albus* profilin is an allergen of 15 kDa and was shown to be IgE-reactive. Nine subjects with a case history of food allergy either to lupine (two, one of them proven by a food challenge, and one additional individual with strongly suspected lupine allergy) and/or to peanut with a suspected lupine allergy/lupine sensitization (seven out of nine), and one individual sensitized to both, lupine and peanut (Figure 3A, Table 1) were tested IgE-positive by immunoblot analysis with the recombinant profilin.

### 3.3. Isolation and Purification of a Non-Specific Lipid Transfer Protein from L. angustifolius Seeds

After the detection of an LMW protein of nearly 10 kDa in immunoblot with an acidic extract of *L. angustifolius* and *L. luteus*, this protein was isolated and further purified via size exclusion chromatography and ion-exchange chromatography from *L. angustifolius* extract and investigated for IgE-reactivity (Figure 4A,B). N-terminal sequencing and BLAST revealed that it was an LTP. The database was queried in January 2017 after the LTP sequence was published in December 2016. The LTP was purified and, after MS/MS analysis, identified as such. However, it was shown that the LTP was unintentionally co-purified with a cysteine proteinase inhibitor, both at approximately 11 kDa, and, therefore, indistinguishable from each other. This could not be seen in the SDS-PAGE under reducing conditions. After the existence of cysteine proteinase inhibitor had become known, SDS-PAGE was performed under non-reducing conditions, where both proteins could be differentiated (Figure 4C, Appendix A). Under reducing conditions, the patient IgE was directed against the 11 kDa “protein band”, whereas the immunoblot under non-reducing conditions shows solely IgE-reactivity to the LTP (16 kDa band) and not to the cysteine proteinase inhibitor at 11 kDa (Figure 4D).

MS-analysis revealed the following sequences (similarity calculated in %). XP_019446786.1:

Lane 3: ITCGQVTANLAQCLNYLRSGGAVPAPCCNGIKNILNLAKTTPDRRTACNCLKAAAANTPGLNPSNAGSLPGKCGVNIPYKISTSTNCASIK: 93.4%

Lane 4: ITCGQVTANLAQCLNYLRSGGAVPAPCCNGIKNILNLAKTTPDRRTACNCLKAAAANTPGLNPSNAGSLPGKCGVNIPYKISTSTNCASIK: 33.3%

Investigation for glycosylation of the LTP from *L. angustifolius* was not performed in detail. However, one potential glycosylation at position 87 (NPSN, analyzed by http://www.cbs.dtu.dk/services/NetNGlyc/) exists. This was not verified by mass spectrometric analysis (meaning AAAANTPGLNPSNAGSLPGK was identified without glycosylation). Protein sequence (complete) was: MAGIVKLACAVLICMVVVSAPLTKAITCGQVTANLAQCLNYLRSGGAVPAPCCNGIKNILNLAKTTPDRRTACNCLKAAAANTPGLNPSNAGSLPGKCGVNIPYKISTSTNCASIK (XP_019446786.1). The N-terminal sequencing revealed the following N-terminus ITXGQVTANLAQ, which was confirmed by LC–MS/MS, and the sequence coverage of the expected full-length protein was 100% (see Appendix A).

#### *Lupinus angustifolius* and *L. luteus* LTP Allergenicity

The investigation for the allergenicity of the newly found LTPs in immunoblot analysis revealed that eight out of 17 patients with lupine allergy and/or peanut allergy and a polysensitized individual were IgE-positive for lupine LTPs.

Three of these individuals had undergone an oral provocation test with lupine flour and developed symptoms [1], and all three were IgE-positive to *Lupinus angustifolius-*LTP. The polysensitized individual was also LTP-IgE-positive. The *L. angustifolius* LTP was submitted to and accepted by the WHO/IUIS allergen nomenclature subcommittee in 2019 as Lup an 3 (Table 2).

## 4. Discussion

Molecular allergology based on many individual allergens belonging to only a few concise protein families has revolutionized the understanding of the pathomechanism of allergies and allergy diagnostic procedures considerably. However, there are still important gaps in diagnostic test sensitivity and specificity [20,21]. Several examples outline the importance of the inclusion of single allergens of clinically relevant allergen sources into diagnostic tests either as component-resolved diagnostics or as additional ingredients in the whole extracts used for diagnostic tests (“spiking” of the extracts with single allergens). Many single allergens have already gone into routine allergy diagnostic tests; however, lupine species are not among them. In vitro tests for lupine allergy presently are based only on lupine seed extract of one species. This is critical because lupine is an upcoming relevant food, and allergy to lupine can be severe, which is why allergists should be prepared and diagnostic procedures updated considerably.

The study presented here is, to the best of our knowledge, the first experimental comparative investigation on potential allergens of all three relevant *Lupinus* species and shows qualitative and quantitative species-associated differences in the protein content. In addition, we achieved for the first time the detection of three single LMW lupine proteins as new allergens, one pan allergen, the profilin of *L. albus,* and two LTPs, and thereby potential marker allergens for severe reactions, a non-specific (ns) LTP of *L. angustifolius* and an ns-LTP of *L. luteus.* The latter could not be submitted as a new allergen to the WHO/IUIS allergen nomenclature subcommittee as yet because our results could not be confirmed with a published sequence at that time and are presently being re-evaluated.

Having single allergens available for routine diagnostic tests allows for the identification of the primary sensitizing food (primary or pollen-associated lupine allergy), the detection of potential cross-reactivity, an increase of in vitro test sensitivity in cases where the extract-based diagnostic test lacks single allergens, and the identification of patients at risk to suffer from severe reactions.

The reactivity of patients to lupine profilin provides evidence for a pollen-induced sensitization to lupine. In general, IgE to profilins are associated with mild symptoms in pollen allergy but can be severe in pollen-associated food allergy [22].

In contrast, severe reactions are mostly associated with storage proteins—which is also hypothesized for lupine—but there are increasing cases not only in the Mediterranean but also in Central and Northern Europe, where severe reactions are associated with ns-LTPs [23]. LTPs are small, lipophilic proteins (91 to 95 amino acids), eight cysteines forming disulfide bridges, basic isoelectric point, α-helical structure [24], an altogether stable structure that is resistant to heat and digestion. They are ubiquitous in the plant kingdom, and investigations regarding potential cross-reactivity revealed that some show a strong structural similarity even when part of plants with a distant taxonomic relationship [25]. This is particularly relevant for a clinical phenomenon called the LTP syndrome, where patients characteristically react to LTPs from phylogenetically different plant food sources [26,27]. Although there is still a clinically silent sensitization to be considered, there are observations that whenever a food-allergic patient has IgE against more than five LTPs from different food sources, there is a risk of developing severe reactions [28]. In addition, there are data on endoluminal food allergy (gastrointestinal symptoms only) as a sequel to ingestion of LTP-containing food [28,29,30]. As can be imagined, these isolated gastrointestinal symptoms are very often not classified as allergic, except an experienced allergist elucidates this connection and supports it with plausible results of allergy diagnostic tests. These are important clinical observations that strongly speak in favor of a broadening of allergen panels to investigate for sensitization patterns that allow the correct phenotyping and—in case marker allergens for the severity of a reaction are involved—an adequate risk evaluation and management. The first LTP, fully characterized as an allergen, was Pru p 3 [31]. Presently, this allergen, which is already part of routine allergy tests, is being used as representative LTP in case a clinician suspects LTP-association of a severe allergic reaction.

The lupine proteins that have been documented so far by the WHO/IUIS allergen nomenclature subcommittee as allergens [19] belong to different lupine species. Single allergens belonging to one protein family have not been isolated from all three lupine species in parallel yet. It is the same in our study. According to our own experience, this is due to methodical difficulties in protein isolation and purification, maybe even due to species-specific differences. In our comparison experiments, we detected a high degree of molecular diversity between the three lupine species, which partly was mirrored by an immunological diversity in so far as patients had different IgE-reactivity to different lupine species. Unfortunately, the 2D-PAGE experiment could not be performed with patient sera due to the lack of sera volumes. Therefore, the question remains as to whether a patient allergic to one lupine species may tolerate another.

After we had purified two lupine allergens, we tested more patients and used these allergens to identify the culprit food in patients with no unambiguous evidence regarding the cause of their food-associated (severe) symptoms. One source of potentially severe lupine allergy is a pre-existing peanut allergy. In our study, we also included peanut-allergic individuals who mostly showed lupine sensitization, but some also suffered from lupine allergy.

All in all, not many lupine-allergic patients were admitted to our outpatient clinic, and—although this is a multicenter study—not many could be included in this investigation. Most of them were oblivious to their lupine allergy and did not even know that lupine is a food, although it must be declared on ingredient lists [32]. Therefore, we believe that there is a huge number of unrecorded allergic and even anaphylactic cases based on lupine, which are most probably documented as idiopathic anaphylaxis.

We think that only after the allergen profiles of foods with high anaphylactic potential have been elucidated, a correct diagnosis accompanied by correctly proposed prophylactic measures can be made. Apart from Lup an 1, ours are the only two other new lupine allergens accepted by the WHO/IUIS allergen nomenclature subcommittee presently as Lup an 3 and Lup a 5. Lup a 4 and Lup l 4 (the Bet v 1-homolog) have already been described and documented in Allergome only so that, in general, some of the most relevant allergen families represented in lupine could be detected in case these allergens went into routine diagnostics. Some research still must be done on the purification of lupine storage proteins, as they are most probably associated with severe reactions and also must be included in routine diagnostic measures. Although building slowly, there will be component-resolved diagnostics for lupine allergy and anaphylaxis in the near future.

## Figures and Tables

**Figure 1 nutrients-13-00409-f001:**
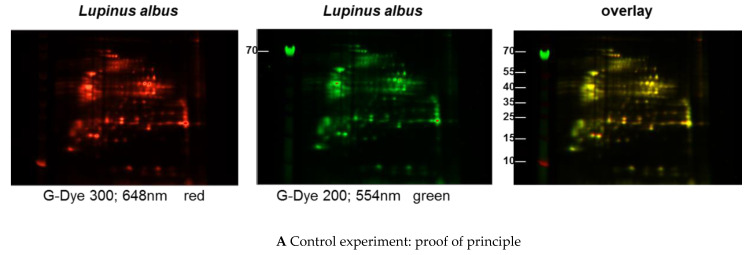
(**A**–**C**) Protein extracts obtained via alkaline extraction of flour from all three lupine species were studied by 2D-DIGE. (**A**) The control experiment shows the complete identity of the differently dyed samples of an alkaline extract of *Lupinus albus* by turning to yellow after having overlaid the two complementary colors, red and green. (**B**) By comparison of extracts from the three lupine species by 2D fluorescence difference gel electrophoresis (2D-DIGE), it becomes evident that there is no strong identity. On the contrary, the dominance of the single colors in the overlay speaks in favor of a broad molecular diversity of the dyed proteins in the different extracts. Whether the diversity is mirrored by immunological diversity was part of the subsequent investigations. (**C**) When comparing peanut extract with the extracts from different lupine species, there are only a few yellow areas. In addition, the differences between the lupine species become evident in this experiment as well, as there are different distributions of proteins colored green (lupine) when overlaid with peanut proteins dyed red.

**Figure 2 nutrients-13-00409-f002:**
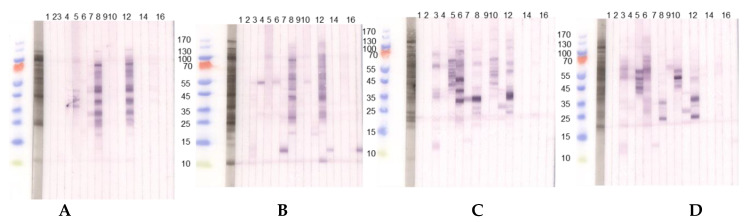
Immunoblots with acidic extracts (0.1 M CH_3_COONH_4_, pH 5.0), TTBS blocking and dilution of sera 1:10 with two different lupine species, (**A**) *L. angustifolius* and (**B**) *L. luteus* chosen as examples. (1) Tris control; (2) negative control serum; (3) lupine and peanut allergy (P 6); (4) legume allergy, suspected lupine allergy, lupine-sensitized (P 2); (5) peanut allergy, suspected lupine allergy, lupine-sensitized (P 3); (6) lupine allergy (P 23); (7) lupine allergy (P 10); (8) peanut allergy, lupine-sensitized (P 24); (9) lupine and peanut allergy (P 25); (10) lupine and peanut allergy (P 26); (11) peanut allergy, suspected lupine allergy (P 27); (12) peanut allergy, lupine-sensitized (P 4); (13) peanut allergy, lupine-sensitized (P 13); (14) lupine allergy (P 29); (15) peanut allergy and suspected lupine allergy (P 28); (16) suspected lupine allergy, lupine-sensitized (P 14); (17) peanut allergy, lupine-sensitized (P 12). (**C**,**D**) Immunoblots with alkaline extracts (0.2 M NH_4_HCO_3_, pH 8.0) of (**C**) *L. angustifolius* and (**D**) *L. luteus* with sera from the same patients. For sera from some individuals (P 4, P 10, P 12, P 13, P 14) differences regarding the IgE-reactivity to different lupine species are detectable. Particularly, sera from P 10, P 12, and P 13 showed reactivity to one LMW protein in the *L. luteus* extract, which we decided to work upon further since the reactivity was dominant when compared to weak or missing reactivity to other proteins in the extract. (P-code corresponds with Table 1.).

**Figure 3 nutrients-13-00409-f003:**
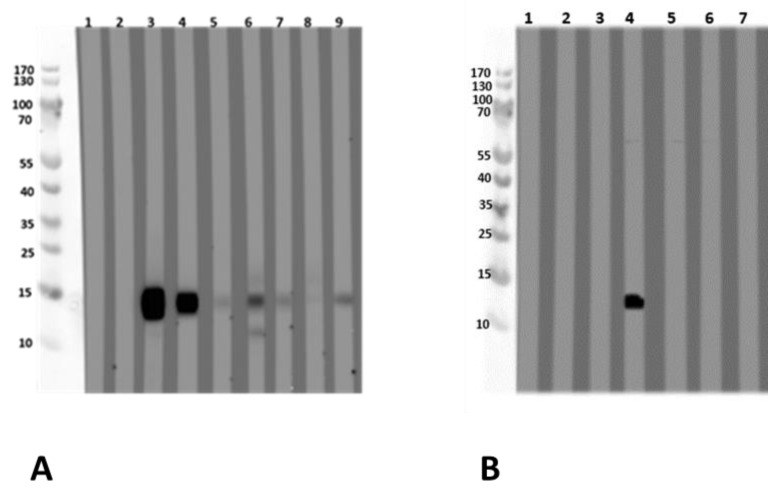
Immunoblot with (**A**) recombinant *L. albus* profilin and (**B**) with natural *L. albus* extract. (**A**) (1) Tris (buffer negative control) (2) negative serum; (3) peanut allergy, lupine-sensitized (P 11); (4) peanut allergy, lupine-sensitized (P 12); (5) suspected lupine allergy (P 14); (6) peanut allergy, lupine-sensitized (P 13); (7) peanut and lupine allergy (P 15); (8) peanut allergy, lupine-sensitized (P 16); (9) peanut allergy, lupine-sensitized (P 17). (**B**) The immunoblot was designed as an inhibition assay using natural *L. albus* extract and sera pre-incubated with different concentrations of recombinant *L. albus* profilin using one profilin-reactive serum (4). (1) Tris, (2) negative serum, (3) negative serum + 50 µg rProfilin, (4) serum 4 (P 12), (5) serum 4 + 5 µg rProfilin, (6) serum 4 + 50 µg rProfilin, (7) serum 4 + 500 µg lupine extract. Blocking was performed with Synblock; serum dilution was 1:20, dilution of the HRP-conjugated mouse-anti-human IgE Fc-antibody was 1:5000 (SouthernBiotech, Birmingham, AL, USA). Immunologically, the recombinant profilin performed similarly to the natural allergen. P-code refers to Table 1.

**Figure 4 nutrients-13-00409-f004:**
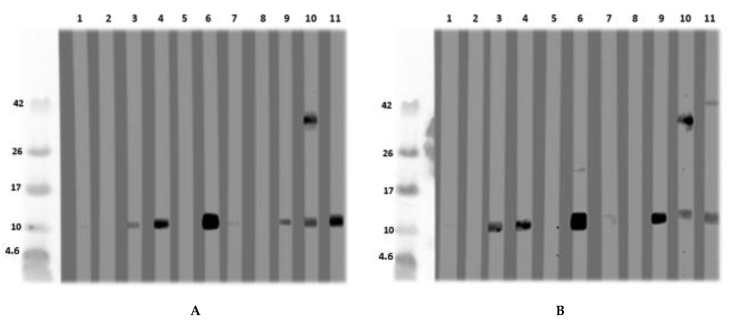
(**A**) Immunoblot with naturally purified *L. luteus* LTP; (**B**) with naturally purified *L. angustifolius* LTP, using in a first step Synblock, in a second step anti-human-IgG 1:500. The sera were diluted 1:10. (1) Tris control; (2) negative control serum; (3) lupine and peanut allergy (P 6); (4) peanut allergy, lupine-sensitized (P 9); (5) peanut allergy, lupine-sensitized (P 5); (6) peanut allergy, lupine-sensitized (P 4); (7) peanut allergy and suspected lupine allergy (P 3); (8) peanut and suspected lupine allergy (P 2); (9) peanut and lupine sensitization (P 1); (10) lupine allergy and peanut sensitization (P 8); (11) lupine and peanut allergy (P 7). (**C**) SDS-PAGE for MS-analysis: *L. angustifolius* LTP (1) and *L. luteus* LTP (2) under reducing conditions. *L. angustifolius* LTP (3) and *L. luteus* LTP (4) under non-reducing conditions. (**D**) Immunoblot with natural *L. angustifolius* LTP (non-reducing conditions) (1) Tris (2nd antibody control) (2) negative serum I (3) negative serum II (4) negative serum III (5) peanut allergy and lupine-sensitized (P 4); (6) peanut allergy and lupine-sensitized (P 9); (7) lupine and peanut allergy (P 6). (P-codes refer to Table 1).

**Table 1 nutrients-13-00409-t001:** Characterization of the patients investigated for IgE-reactivity with lupine extracts and new single allergens.

P-Code	Gender	Age	Peanut Allergy	Peanut Sensitization	Lupine Allergy	Lupine Sensitization	BLOT					CAP					
							*L. luteus* (Whole Extract)	*L. angustifolius* (Whole Extract)	*L. luteus* LTP	*L. angustifolius* LTP	*L. albus* Profilin	Total-IgE	Lupine Seed	Peanut Extract	Ara h 2	Ara h 8	Ara h 9
**P 1**	M	35		x		x	+	+	+	+	n. d.	>5000	n. d.	84.7	n. d.	n. d.	n. d.
**P 2**	F	25	x	x	(x)	x	+	+	−	−	n. d.	110	4.75	n. d.	n. d.	n. d.	n. d.
**P 3**	F	24	x	x	(x)	x	+	+	(+)	(+)	n. d.	n. d.	5.54	n. d.	>100	0	n. d.
**P 4**	M	76	x	x	?	x	+	+	+	+	n. d.	1053	n. d.	n. d.	n. d.	n. d.	n. d.
**P 5**	F	27	x	x	?	x	+	+	−	−	n. d.	1828	n. d.	47.5	28.9	11.4	0.06
**P 6**	F	63	x	x	x	x	+	+	+	+	n. d.	199	26.6	1.27	n. d.	n. d.	5.83
**P 7**	F	24	x	x	x	x	+	+	+	+	n. d.	n. d.	6.5	>100	n. d.	n. d.	n. d.
**P 8**	F	42		x	x	x	+	+	+	+	n. d.	>5000	4.9	18	0	50	0
**P 9**	M	15	x	x	?	x	+	+	+	+	n. d.	705.5	n. d.	21.3	3.21	1.34	3.7
**P 10**	M	63			x	x					+		0.75	n. d.	n. d.	n. d.	n. d.
**P 11**	F	30	x	x	?	x	+	+	−	−	+	1696	n. d.	80.7	n. d.	>100	n. d.
**P 12**	F	57	x	x		x	+	−	−	−	+	357	3.69	4.2	<0.1	6.2	0
**P 13**	F	27	x	x		x	+	−	−	−	+	1324	0.71	13.7	8.56	4.7	<0.01
**P 14**	F	50			(x)	x	+	−	−	−	+	n. d.	0.64	n. d.	n. d.	n. d.	n. d.
**P 15**	F	38	x	x	x		−	−	−	−	+	n. d.	<0.35	0.8	n. d.	n. d.	n. d.
**P 16**	M	24	x	x		x	+	+	n. d.	n. d.	+	208	1.28	39.8	36.4	0.67	0
**P 17**	M	28	x	x		x	+	+	n. d.	n. d.	+	94.3	0	0.74	0	0	0
**P 18**	M	24	x	x		x	+	+	−	−	+	1416	11	88.4	34.1	9.93	0.15
**P 19**	M	33	x	x		x	+	+	n. d.	n. d.	n. d.	595.6	5.55	75.9	51	7.95	0
**P 21**	M	10	x	x			n. d.	n. d.	+	n. d.	n. d.	531	n. d.	52.1	18.2	0.17	36.9
**P 22**	M	43		x		x	+	+	n. d.	n. d.	+	83.5	<0.01	0.22	<0.01	1.65	0.30
**P 23**	F	67		x	x	x	+	+	−	−	−	116	42.9	0	n. d.	0.46	n. d.
**P 24**	M	32	x	x		x	+	+	n. d.	n. d.	−	n. d.	7.02	37.1	n. d.	n. d.	n. d.
**P 25**	F	9	x	x	x	x	+	(+)	n. d.	n. d.	−	n. d.	11.2	>100	n. d.	n. d.	n. d.
**P 26**	F	39	x	x	x	x	+	+	n. d.	n. d.	−	n. d.	1.58	n. d.	n. d.	n. d.	n. d.
**P 27**	F	34	x	x	(x)		+	+	n. d.	n. d.	−	137	<0.12	2.76	<0.1	0.29	<0.01
**P 28**	F	32	x		(x)		(+)	(+)	n. d.	n. d.	−	319	16.7	n. d.	67.6	0	n. d.
**P 29**	F	51			x		−	−	n. d.	n. d.	n. d.	67.3	0	0	0	0	0
**P 30**	F	26	x	x	x	x	+	+	n. d.	n. d.	−	1023	n.d.	>100	>100	n. d.	<0.01
**P 31**	M	30	x	x			−	−	n. d.	n. d.	−	936	n. d.	n. d.	0.04	n. d.	n. d.

M: male; F: female; x: convincing history with or without provocation test; (x) strongly suspected lupine allergy; ?: not knowingly consumed; n. d.: not done. Due to the small volumes of some sera, not all parameters could be investigated for every patient.

**Table 2 nutrients-13-00409-t002:** Synopsis of single allergens in lupine species compared to peanut allergens.

Plant Food Allergens (Protein Families)	Peanut Allergen (*Arachis hypogaea)* Ara h x	MW [kDa]	Lupine Allergen (*Lupinus angustifolius):* Lup an x *(Lupinus albus)*: Lup a x	MW [kDa]
Vicilin-type storage protein; 7S globulin	Ara h 1 (IUIS)	64	Lup an 1 β-conglutin (IUIS) *Lup a 1*	55–61
Conglutin-like storage protein; 2S albumin	Ara h 2 (IUIS)	17	δ-conglutin *Lup a δ-conglutin* *Lup an δ-conglutin*	2 4
Legumin-type; 11S globulin	Ara h 3 (IUIS)	60	α-conglutin Lup a α-conglutin Lup an α-conglutin	43
Profilin	Ara h 5 (IUIS)	15	Lup a 5 (IUIS)	15
Conglutin; 2S albumin	Ara h 6 (IUIS)	15		n.a.
Conglutin; 2S albumin	Ara h 7 (IUIS)	15	*Lup a γ-conglutin*? *Lup an γ-conglutin*?	n.a
PR-10 Bet v 1-super family	Ara h 8 (IUIS)	17	*Lup a 4* *Lup l 4*	16.5 17
Non-specific lipid transfer protein	Ara h 9 (IUIS) Ara h 16 (IUIS) Ara h 17 (IUIS)	9.8 8.5 11	Lup an 3 (IUIS) *L. luteus*	11 11

[3], modified; [18]. Italics: new, but not documented/accepted by the WHO/IUIS Allergen Nomenclature Subcommittee [19]; MW for IUIS accepted allergens are given as MW (SDS-PAGE); n.a.: not available.

## Data Availability

Data is contained within the article and the Appendix A.

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
