# Peer review of "Identification and Purification of Novel Low-Molecular-Weight Lupine Allergens as Components for Personalized Diagnostics"

_nutrients, 2021, doi:10.3390/nu13020409_

Round 1
Reviewer 1 Report
The entire manuscript is interesting and well described. Crucial information about food allergy is included in the introduction, for example, about cross-reactions between peanut proteins (eg Ara h 1) and lupine allergens. I think that the manuscript should be read not only by medical personnel, but also by every patient with food allergy.
Before publishing the work, I have some questions:
- Where did the lupine for the test flour come from?
- How many volunteers took part in the study? This should be described in the methodology, not only in the table.
- Was there an allergy free control group?
- Why did only three participants take part in the double-blind, placebo-controlled trial? How and where were they selected?
Minor corrections:
- Line 130 – missing dot ('.')
- remove spaces between ‘37 °C’ , such as on line 215, and others
- Line 233 – missing ‘C’ after ‘37°’
- Line 622- ‘We, therefore, believe that there is a huge number of unrecorded allergic and 622 even anaphylactic cases based on lupine which are most probably documented as idiopathic 623 anaphylaxis’ change into ‘Therefore, we believe…’
Author Response
Point-by-point Reply to the Reviewers Comments
Comments of Reviewer 1
Before publishing the work, I have some questions:
- Where did the lupine for the test flour come from?
Response: Seeds of Lupinus albus (var. Feodora), was provided by Saaten-Union GmbH, Isernhagen, Germany
Seeds of Lupinus angustifolius (var. Boregine) was provided by Saatszucht Steinach, Steinach, Germany.
Lupinus luteus (var. Juno ZS) was provided by Feldsaaten Freudenberger GmbH & Co KG, Krefeld, Germany
This information has been added to Material and Methods section 2.2. Lines 130-133
- How many volunteers took part in the study? This should be described in the methodology, not only in the table.
Response: lines 111- 115. This clarification has been added to the information given under “Study Group”. It now reads: “Patients with lupine allergy alone (including one with strongly suspected allergy to lupine-containing foods) (n=5), 10 patients with peanut allergy and lupine allergy (including four with strongly suspected lupine allergy), 12 patients with peanut allergy and lupine sensitization, two patients with peanut allergy without proven lupine sensitization, two patients with lupine and peanut sensitization only, and one non-allergic individual, whose serum served as negative control were recruited during clinical….”
- Was there an allergy free control group?
Response: So far, only one non-allergic individual was included, whose serum served as negative control for the IgE-detection method. The aim of the study was to identify new lupine allergens. In order to achieve this goal, predominantly allergic individuals were recruited. However, we thank the reviewer for this comment and shall include allergy free individuals in our next steps of the project.
- Why did only three participants take part in the double-blind, placebo-controlled trial? How and where were they selected?
Response: These three patients had been recruited and challenged in Utrecht as documented in the Material and Methods section under “Study Group” in a trial published by Peeters et al., 2009: 1. Peeters, K.A.; Koppelman, S.J.; Penninks, A.H.; Lebens, A.; Bruijnzeel-Koomen, C.A.; Hefle, S.L.; Taylor, S.L.; van Hoffen, E.; Knulst, A.C. Clinical relevance of sensitization to lupine in peanut-sensitized adults. Allergy. 2009, 64, 549-555. The sera from lupine challenge positive patients from this study were provided by A. Knulst (senior author) to investigate them for IgE to new allergens in our study. The study was referenced as (1) in the section study group as well as in the respective result section.
It now reads: “Three had undergone double blind placebo controlled lupine challenge in Utrecht with positive result and were included in this study [1].” Lines 125-126. I also provided the reference in the respective result section, line 556
Minor corrections:
- Line 130 – missing dot ('.')
Response: Done
- remove spaces between ‘37 °C’ , such as on line 215, and others
Response: Done
- Line 233 – missing ‘C’ after ‘37°’
Response: Done
- Line 622- ‘We, therefore, believe that there is a huge number of unrecorded allergic and 622 even anaphylactic cases based on lupine which are most probably documented as idiopathic 623 anaphylaxis’ change into ‘Therefore, we believe…’
Response: Done
Reviewer 2 Report
The paper by Jappe et al describes the characterization of 3 new allergens from Lupine species. Lupine proteins are currently found in commercial products as an additive, and many consumers are unaware of the ingredients and/or potential allergen content. Hence it is important to characterize potential allergens and understand cross-reactivity with other allergens. Jappe and colleagues carefully analyzed 3 lupine species for allergens and probed with two different extraction procedures using primarily different pH. 2D gels revealed major differences in the proteome among the lupine species and differences from peanuts. Immunological differences were probed with sera from 15 patients allergic to peanuts and/or lupine allergic or suspected allergic patients. Ultimately, they identified a profilin Lup a 5, and an LTP Lup an 3, which belong to common food and plant allergen families. The authors suspect another LTP allergen but did not get confirmed sequence information from L. luteus. Overall the data should be useful in molecular allergology.
The authors should be commended for writing a clear paper with plenty of technical details for reproducibility. The authors should also be commended for submitting allergen names to WHO/IUIS for proper nomenclature.
There are no major issues with this paper.
The only minor issue is that Table 2 appears to be duplicated in the supplementary material, in a slightly different format.
Author Response
Point-by-point Reply to the Reviewers Comments
Comments of Reviewer 2
Comments and Suggestions for Authors
Overall the data should be useful in molecular allergology.
The authors should be commended for writing a clear paper with plenty of technical details for reproducibility. The authors should also be commended for submitting allergen names to WHO/IUIS for proper nomenclature.
There are no major issues with this paper.
Response: The authors wish to thank the reviewer for his kind and encouraging comments.
The only minor issue is that Table 2 appears to be duplicated in the supplementary material, in a slightly different format.
Response: The reviewer is correct. Table 2 should not be part of the supplementary figures but of the manuscript instead. However, the submission process somehow did not allow to upload the second table of the manuscript. This has already been realized and corrected by the production team.
I omitted Table 2 from the Supplementary material.